# Prevalence of unqualified sources of antimalarial drug prescription for children under the age of five: A study in 19 low- and middle-income countries

Md Sabbir Hossain[1], Talha Sheikh Ahmed[2], Mohammad Anamul Haque[1], Muhammad Abdul Baker Chowdhury[1], Md Jamal Uddin[1,3]*

1 Biostatistics, Epidemiology and Public Health Research Team, Department of Statistics, Shahjalal University of Science and Technology, Sylhet, Bangladesh, 2 Department of Geography and Environment, Shahjalal University of Science and Technology, Sylhet, Bangladesh, 3 Department of General Educational and Development, Daffodil International University, Dhaka, Bangladesh

* jamal-sta@sust.edu

**Data Availability Statement:** All relevant data are within the paper and its Supporting Information files.

## Abstract

### Background

Antimalarial drug resistance poses a severe danger to global health. In Low- and Middle-Income Countries (LMICs), there is a lack of reliable information on antimalarial prescriptions for recent malarial fever in children under five. Our study aims to determine the prevalence of unqualified sources of antimalarial drug prescription for children under the age of five in 19 low- and middle-income countries.

### Methods

We performed a cross-sectional study of the Malaria Indicator Survey (MIS) datasets (n = 106265) across 19 LMICs. The recent MIS datasets were used, and the study only included children under five who had taken an antimalarial drug for a recent malarial fever. The outcome variable was classified into two distinct categories: those who had taken antimalarial drugs for malarial fever from qualified sources and those who did not.

### Findings

Among LMICs, we found that 87.1% of children under five received an antimalarial prescription from unqualified sources who had recently experienced malarial fever. In several LMICs (Tanzania, Nigeria, and Ghana), a substantial portion of recent antimalarial prescriptions for malaria was taken from unqualified sources (about 60%). Some LMICs (Guinea (31.8%), Mali (31.3%), Nigeria (20.4%), Kenya (2.6%), and Senegal (2.7%)) had low rates of antimalarial drug consumption even though children under five received a high percentage of antimalarial prescriptions from qualified sources for a recent malarial fever. Living in rural areas, having mothers with higher education, and having parents with more wealth were frequently

**Funding:** The author(s) received no specific funding for this work.

**Competing interests:** The authors have declared that no competing interests exist.

taken antimalarial from qualified sources for recent malarial fever in children under five across the LMICs.

## Interpretation

The study draws attention to the importance of national and local level preventative strategies across the LMICs to restrict antimalarial drug consumption. This is because antimalarial prescriptions from unqualified sources for recent malarial fever in children under five were shockingly high in most LMICs and had high rates of unqualified prescriptions in certain other LMICs.

## Introduction

Malaria is a potentially fatal disease that may be prevented and treated. It is spread to people by specific mosquito species. According to the World Health Organization's (WHO) worldwide malaria program, there were 619,000 malaria-related deaths worldwide in 2020, a modest decrease from the anticipated 247 million cases in the previous year [1]. Due to interruptions in crucial malaria interventions during the pandemic of COVID-19, there were 63,000 deaths between 2019 and 2021 [2].

In Sub-Saharan African (SSA) nations, it is a significant source of infectious diseases and mortality. In the WHO African Region, malaria fatalities climbed from 544,000 to 599,000 during 2019 and 2020, while anticipated cases rose from 218 million to 232 million. Deaths fell to 593,000 in 2021, while cases rose to 234 million [3]. 38.4% of all malaria-related deaths in children under 5 years old occurred in Nigeria [3]. According to estimates, children under the age of five account for about two-thirds of deaths of which SSA countries result in 90% of cases and 97% of deaths related to malaria [1, 4].

Efforts to combat malaria involve adopting new diagnostic techniques, medications, and control strategies. Understanding the elements that affect patient medication-seeking patterns and choices to purchase anti-malarial drugs in the private sector still has gaps [5, 6]. It is crucial to confirm a malaria diagnosis before administering treatment. The World Health Organization (WHO) has advocated for parasitological confirmation using microscopy or rapid diagnostic tests (RDT) in all suspected cases since 2010 [7]. Studies have revealed that in SSA countries, rather than using public health facilities, most families initially turn to the retail industry for the treatment of mild febrile diseases [8, 9]. It is more frequent due to limited access to health services [10–12].

Correct diagnosis of the illness is crucial to attaining effective malaria outcomes [13, 14]. Incorrectly treating non-malarial fevers with burdensome artemisinin-based combination therapy (ACT) due to medication presumptive treatment will promote the establishment of parasite resistance and risk the lives of patients and the standard of care [15]. A systematic review revealed that obtaining anti-malarial medications did not necessarily require an accurate malaria diagnosis or a prescription. Surprisingly, having a drug prescription was not a strong predictor of purchasing ACT, even though a higher proportion of ACT drugs was obtained with a prescription in Kenya, Nigeria, and Tanzania [16]. In low-middle income countries (LMICs), where the privately owned sector supplies patients with around 60% of their malaria drugs, these findings are crucial for the formulation of national malaria diagnosis and treatment policies [17, 18]. They demonstrate that patients can obtain the WHO-recommended ACT treatment without utilizing public healthcare facilities, an accurate diagnosis of

malaria, or a prescription for an anti-malarial medication. Moreover, expanding healthcare access in both public and private sectors is expected to improve fever diagnosis and treatment, irrespective of whether the fever is caused by malaria or another illness.

Over the past two decades, substantial global efforts through various policies and interventions have contributed to a commendable 47 percent reduction in mortality rates among children under five from malaria between 2000 and 2019 [19]. Despite this progress, the sobering reality remains that a child under five succumbs to malaria every two minutes, underscoring the urgency to bolster protection for vulnerable populations, particularly children [20]. This persistent threat is exacerbated by the emergence of antimalarial drug resistance, jeopardizing the effectiveness of existing treatment strategies. To assess the adequacy of current policies and ensure the delivery of high-quality malaria diagnosis and treatment, regular performance reviews by National Malaria Programmed are imperative. Furthermore, enhancing treatment outcomes demands a concerted effort to minimize the risk of drug resistance, emphasizing confirmatory diagnostics and ensuring proper dosing and quality of first line antimalarials. In the broader context, the development of improved algorithms for managing patients testing negative for malaria is pivotal for the effective management of acute febrile illnesses. These multifaceted measures are essential for sustaining and advancing the gains made against malaria and fortifying global efforts towards its control and elimination [21–23].

To our best knowledge, no study has been conducted to focus on anti-malarial drug consumption in LMICs settings without diagnosing that an individual has transmission of malaria in children under five. Therefore, the goals of our study are to identify the prevalence of unqualified sources of antimalarial drug prescription for recent malarial fever in children under the age of five. We will also provide a complete picture of antimalarial prescription by region-wise GIS map in each LMICs.

## Methods

### Data source

Using the most recent malaria indicator survey (MIS) data from 2011 to 2021, we conducted a cross-sectional study in 19 malaria-endemic countries. The Monitoring and Evaluation Working Group (MERG) of Roll Back Malaria, an international collaboration intended to coordinate worldwide efforts to control malaria, established the MIS, a household survey. The MIS comprises questionnaires, manuals, and instructions based on information from the Demographic and Health Surveys (DHS). It gathers regional and national data from a representative sample of respondents. The DHS Program co-chairs the MERG Survey and Indicator Guidance Working Committee and has made significant contributions to the design of the MIS package.

The MIS is often scheduled to coincide with the peak period of malaria transmission and is carried out in close to 30 LMICs. The survey contributes to the collection of important data on the incidence of malaria, the usage and ownership of insecticide-treated mosquito nets, and the efficiency of malaria control measures. The Roll Back Malaria Alliance has created a MIS toolkit to assist with the survey's execution. It includes instructions, questions, and manuals, as well as suggested tabulations for data analysis [24].

### Study design

The MIS survey employs a two-stage stratified cluster sampling method. The first step was to choose specific areas or clusters. The second stage entails picking every cluster or enumeration area's (EA) household in a methodical manner. MIS surveys follow a set of standard operating procedures that include sampling, questionnaires, data collection, cleaning, coding, and

analysis to facilitate cross-country comparisons. The respondents gave both verbal and written consent. The institutional review boards of ICF International and the ethics regulatory bodies of the countries for which the study is conducted normally approve MIS as being ethical.

The MIS program collects data on insecticide-treated mosquito net ownership and usage, intermittent prenatal care, blood tests to diagnose fever in children under five, and indoor residual pesticide spraying are among the indications of malaria that are widely recognized. In addition, the MIS gathers information on household demographics and possession of items like indoor plumbing, electricity, bicycles, and radios. The most vulnerable family members, such as young children and pregnant women, can also have their anemia and malaria parasite counts measured as part of the MIS. Those who qualify and provide their agreement provide a few drops of blood, which are then instantly examined on-site for anemia by specially trained interviewers. More than 60 DHS and MIS surveys have tested for anemia so far, while many more MIS surveys and several DHS surveys have tested for malaria prevalence [24].

### Ethics statement

This research harnessed Malaria Indicator Survey (MIS) datasets sourced from the Demographic and Health Surveys program, ensuring rigorous adherence to ethical principles. MIS gained approval from ICF International's institutional review boards and conformed to ethics regulations in the respective countries of survey implementation. Attaining both verbal and written informed consent, MIS provided participants with comprehensive information about survey objectives, data collection methods, potential benefits, and risks. The paramount voluntary nature of participation was emphasized, and inquiries were encouraged. Stringent measures were adopted to uphold data confidentiality and anonymity, as all information underwent de-identification before analysis, eliminating personal identifiers and preserving participants' privacy in line with ethical standards. The MIS survey datasets are freely available for researchers across the world.

### Data gap analysis on antimalarial prescriptions

We found that there is presently inadequate data to evaluate the efficacy of the different treatment cascades for the disease, despite the presence of nationally representative surveys with a focus on malaria, such as the MIS. This implies that malaria surveys may not adequately capture the quality of treatment.

Second, important information that would have improved access to malaria medications was missing in the survey, such as the reasons why patients rejected taking treatment. There was no information available on blood testing for four countries with malaria endemic. Just five of the 25 nations that were a part of the study supplied information on the findings of malaria tests that might help determine the optimum course of treatment for febrile children under five.

Finally, the information may not fully reflect the course of therapy because it was obtained from mothers' experiences with the antimalarial drugs their kids took rather than the real doctor's prescriptions. Another element that is typically missing from surveys is patient history, which should be included because a patient's ability to get treatment for malaria may be impacted by recurrent malaria.

Given these information-based gaps, our findings which are detailed in the following sections estimate antimalarial prescription patterns in areas where the data are the most complete and where reasonable assumptions can be made. This information is crucial for demonstrating whether an antimalarial prescription was given for a recent case of malarial fever in each region across 19 LMICs.

## Data harmonization

The most recent MIS data for 25 LMICs were collected from the website https://dhsprogram.com/. Only children under the age of five were included in our study, hence we used "Children Recode (KR)" datasets to conduct our analysis. Nineteen LIMCs were finally (Angola, Burkina Faso, Burundi, Ghana, Kenya, Liberia, Madagascar, Malawi, Mali, Mozambique, Nigeria, Rwanda, Senegal, Sierra Leone, Tanzania, Togo, Uganda, and Zambia) selected from 25 possible LMICs. These countries were selected because they matched our inclusion criterion and provided recent MIS data. Six LMICs (Cambodia, Cameroon, Ethiopia, Gambia, Namibia, and Zimbabwe) were excluded since there were insufficient datasets and pertinent variables. It was not known how high the non-response rate was in any of the countries. The reason was that the MIS survey reports were not available for all the countries and among those available, not all reports include information on non-response rates.

Unequal unit selection probabilities are one issue that should be considered when analyzing survey datasets. For computing standard errors, sample weights are crucial because they aid in removing bias that might result from disproportionate sampling and the effects of non-response. Consequently, omitting weights from the study might lead to estimates that are heavily biased. To ensure accurate standard error and p-value calculations, sample weights provided by the MIS survey were used in the current investigations.

Statistical software Stata introduced singleton to manage a single PSU in a stratum. Missing data is only one of the many causes that might lead to a single PSU in a stratum [25]. This causes several issues when studying the data, such as the inability to compute standard errors. We utilized singleton (scaled) to handle singleton PSUs in each stratum. For singleton (scaled), we used the average of the variances from the strata with different sample units as a scaling factor for singleton (certainty) for each stratum. For simplicity of interpretation and analysis, all levels of categorical explanatory variables were specified appropriately. We combined the datasets from each country after extracting the study variables.

## Outcome variable

A dichotomous variable with the category "YES = 1/NO = 0" describes the outcome variable "Antimalarial Taken for Malarial Fever from Qualified Sources". The survey asked if the children had recently had a fever or a cough. If the response was yes, we investigated whether they had recently received antimalarial medicine for treatment. Finally, the qualified sources of the prescriptions were investigated [Fig 1]. In contrast, non-qualified sources included pharmacies, stores, churches, traditional healers, marketplaces, drug vendors, friends/relatives, supermarkets, shops, and others. We classified any government hospitals, private hospitals, clinics, NGOs, and public health sectors as qualified sources. While they are specialists in administering medication, pharmacists in LMICs are not permitted to give antimalarials. Our classification of "pharmacy" as an unqualified supplier of antimalarials reflects this.

## Explanatory variables

The MIS survey revealed two categories of explanatory factors: level one (individual-level variables) and level two (community-level variables). Level one included the age and sex of a child, the highest education attainment of a mother, the number of children under the age of 5 in the household, and the household wealth index. Country and place of residence (rural, urban) were level two factors. These variables were recorded to make them pertinent for analysis and interpretation. Detailed descriptions of the study variables are presented in **S1 Table in S1 File**.

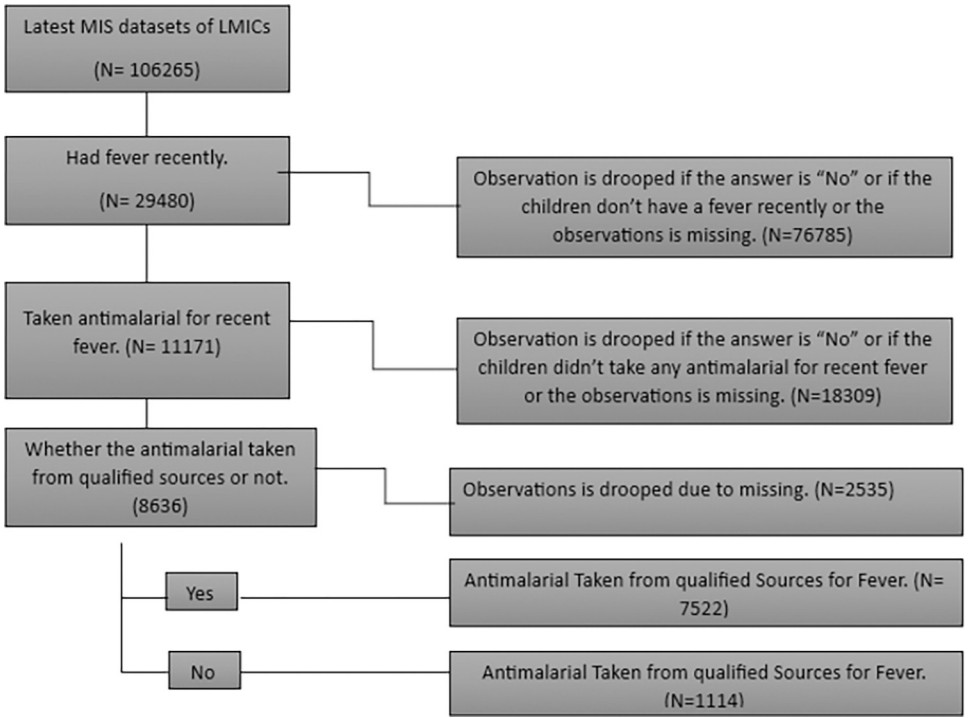

**Fig 1. The outcome variables extraction procedure.**

## Statistical analysis

We calculated various descriptive statistics of qualified antimalarial prescriptions with respect to countries and the explanatory variables. Weighted estimations serve as the foundation for all the presented findings in the tables and figures. To provide a thorough picture of the antimalarial prescriptions from qualified sources in each region of the various LMICs, a geospatial analysis was carried out.

The STROBE Statement Checklist of things that should be included in reports of cross-sectional studies was followed in this study [**S1 Checklist**]. All statistical analyses were performed in Stata version 14 [26] and ArcGIS [27].

## Results

### Descriptives analysis

After combining the MIS datasets from each of the 19 LMICs, we found 106265 children under the age of five. Males make up 54036 of them, or 51%, while females make up 52229, or 49% [**Fig 2**]. In addition, we identified 29,480 (27.74%) children under five who had a recent malarial fever had taken an antimalarial. Among these children, 11171 had taken antimalarial drugs. Ultimately, we determined that 7522 (87.1) % of children under five had recently received antimalarial drugs for malarial fever from qualified sources across the LMICs. [**Fig 1**].

### Country descriptives

Overall, the percentage of antimalarial drugs taken from qualified sources in children under five for recent malarial fever was very high (above 60%) in all LMICs. Among the 19 LMICs, Senegal (100%), Angola (98.7%), Mozambique (98.2%), Burkina Faso (97.2%), and Sierra

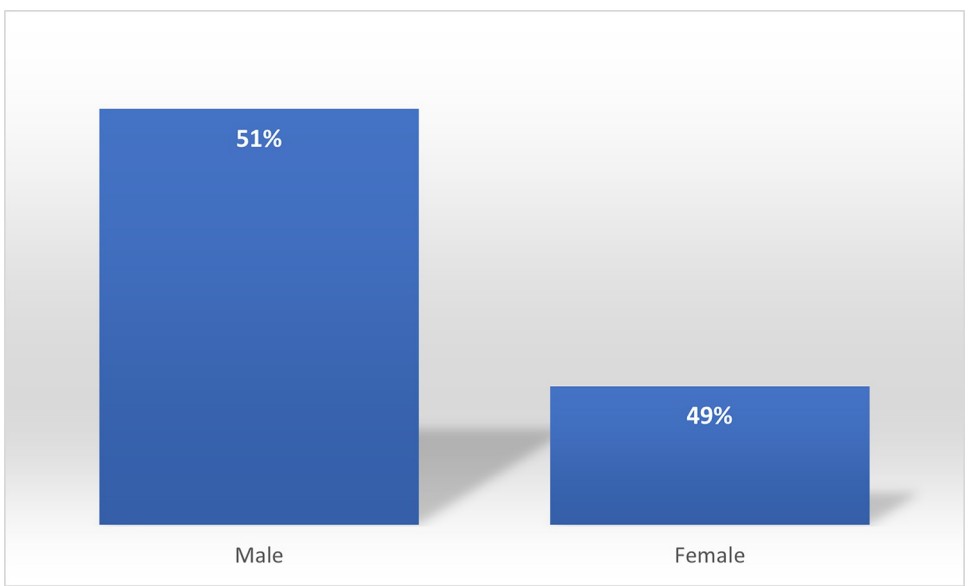

**Fig 2. Gender distribution of children under five in 25 LMICs.**

Leone (96.6%) have the highest percentage of antimalarial prescriptions from qualified sources in children under five for recent malarial fever. Although, Senegal has the highest percentage of antimalarial prescriptions from qualified sources in children under five for recent malarial fever, but the percentage of antimalarial drug consumption was only 2.7%. On the other hand, Tanzania (59.9%), Nigeria (60.0%), Ghana (61.1%), Liberia (75.0%), and Kenya (83.4%) have the lowest percentages of antimalarial drugs from qualified sources in children under five for recent malarial fever. [Table 1] [Fig 3].

**Table 1. Weighted descriptive statistics of antimalarial prescription from qualified sources in children under five for recent malarial fever in 19 LMICs.**

| Country | Had Fever Recently, Weighted N (%) | Antimalarial taken for Malaria, Weighted N (%) | Antimalarial taken from Qualified Sources, Weighted N (%) |
|---|---|---|---|
| Angola | 2632 (34.4) | 706.1 (95.0) | 656.3 (98.7) |
| Burkina Faso | 1176 (20.3) | 600.2 (99.7) | 556.8 (97.4) |
| Burundi | 1806 (35.5) | 455.4 (98.2) | 404.4 (92.1) |
| Ghana | 811.5 (30.1) | 370.2 (99.8) | 197.8 (61.1) |
| Guinea | 909.9 (23.3) | 289.7 (31.8) | 235.8 (91.5) |
| Kenya | 559.2 (17.4) | 14.78 (2.6) | 11.51 (83.4) |
| Liberia | 1011 (40) | 642.2 (97.0) | 446.4 (75.0) |
| Madagascar | 1087 (16.1) | 68.33 (61.7) | 51.36 (87.8) |
| Malawi | 1038 (40.7) | 305.4 (99.8) | 247.7 (92.4) |
| Mali | 2494 (27.3) | 779.7 (31.3) | 684.2 (93.3) |
| Mozambique | 1458 (31.3) | 1458 (100) | 422.3 (98.2) |
| Nigeria | 3917 (36.8) | 799.5 (20.4) | 394.8 (60.0) |
| Rwanda | 884.1 (31.1) | 884.1 (100) | 153.3 (92.0) |
| Senegal | 1654 (30.0) | 44.0(2.7) | 44 (100) |
| Sierra Leone | 1535 (27.1) | 868.2 (99.4) | 763.9 (96.6) |
| Tanzania | 1429 (20.6) | 504.8 (99.3) | 254 (59.9) |
| Togo | 747.5 (24.4) | 225.2(98.4) | 190.1 (94.6) |
| Uganda | 1715 (27.1) | 1074 (98.8) | 883.4 (87.8) |
| Zambia | 2616 (21.4) | 1081 (99.5) | 924.4 (93.2) |

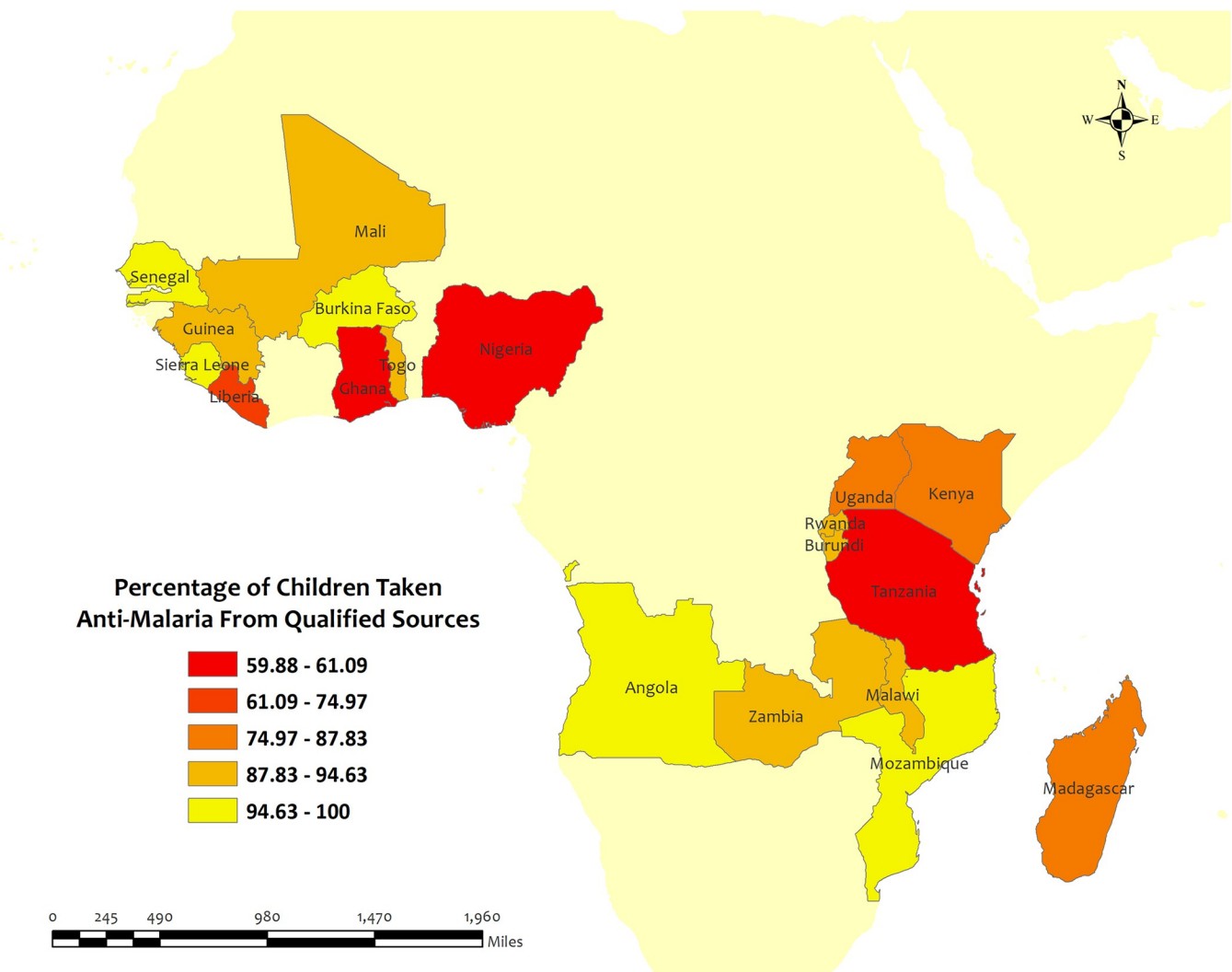

**Fig 3. The overall prevalence of qualified prescription of antimalarial from qualified sources in 19 LMICs.** Here, the darker shades of red indicate unqualified sources of antimalarial. Basemap data provided by ArcGIS (source:https://hub.arcgis.com/datasets/esri::world-countries-generalized/explore?location=76.272878%2C-132.879863%2C12.88).

## Region wise descriptives of five countries with highest qualified prescription

In this section, we presented region-wise prevalence data for the top five countries with the highest antimalarial prescriptions in children under five for recent malarial fever. In Angola, overall antimalarial prescription percentages were high, ranging from 95.5% in Mesoendemic Unstable to 100% in Luanda [**Fig 4**]. Burkina Faso exhibited high overall percentages in all regions, ranging from 95.5% in Hauts-Bassins to 100% in Centre, Centre Est, Centre Nord, Est, Nord, and Sud Ouest [**Fig 5**]. Mozambique recorded high percentages, ranging from 93.9% in Cabo Delgado to 100% in Niassa, Zambezia, Tete, Manica, Sofala, Gaza, Maputo Provincia, and Maputo Cidade [**Fig 6**]. Sierra Leone demonstrated high percentages, ranging from 94.0% in Southern to about 100% in Eastern and North [**Fig 7**]. Senegal reported 100%

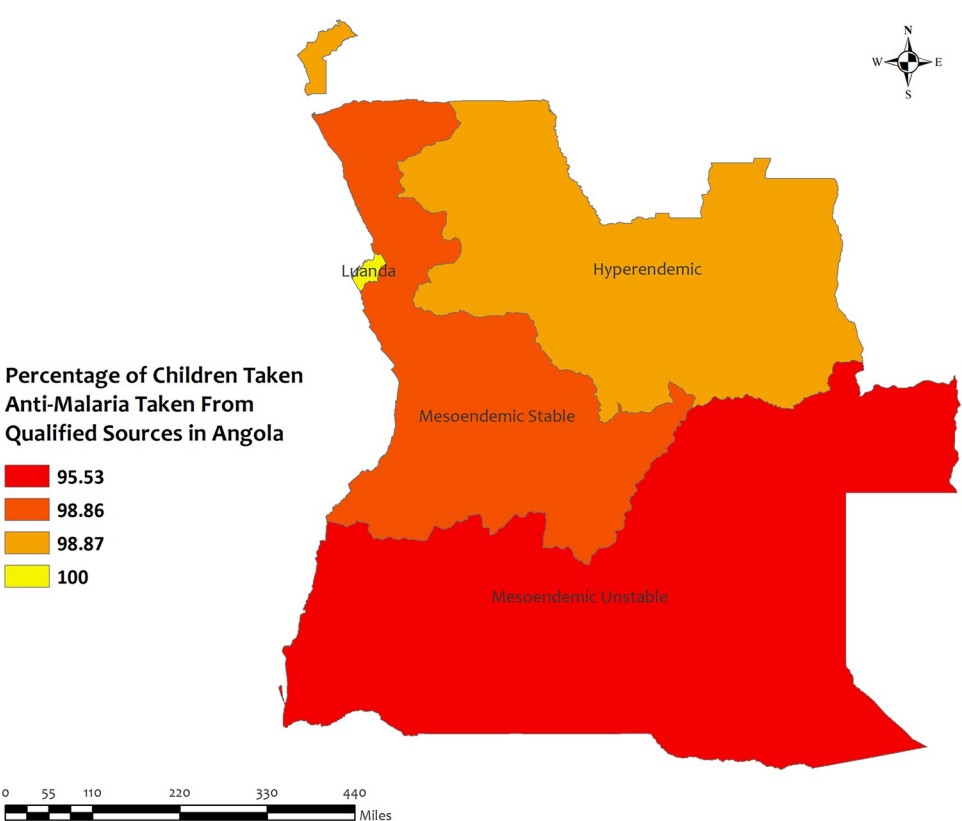

**Fig 4. The prevalence of qualified prescription of antimalarial from qualified sources in Angola.** Here, the darker shades of red indicate unqualified sources of antimalarial. Basemap data provided by ArcGIS (source:https://hub.arcgis.com/datasets/esri::world-countries-generalized/explore?location=76.272878%2C-132.879863%2C12.88).

antimalarial prescriptions in all regions, with varying antimalarial drug consumption percentages, from 0% in Dakar, Kaolack, Thies, and Fatick to 9.7% in Sedhiou [**Fig 8**].

Moreover, antimalarial drug consumption is notably high in Angola, Burkina Faso, Mozambique, Sierra Leone, and Senegal, as emphasized in **S4, S5, S14, S17, and S18 Tables in S1 File**.

## Region wise descriptives of five countries with lowest qualified prescription

In this section, we presented the region-wise prevalence of five countries with the lowest antimalarial prescription from qualified sources for recent malarial fever in children under five.

In Ghana, the overall percentages of antimalarial prescriptions from qualified sources in children under five for recent malarial fever were high in all regions except for Greater Accra. The highest percentages of antimalarial medicine from qualified sources were observed in Upper West (96.6%) and the lowest percentages occurred in the Greater Accra (23.1%) and Ashanti (33.5%) regions [**Fig 9**]. However, the prevalence of antimalarial drug consumption was about 100% in all regions except for the Western (98.8%) region [**S7 Table in S1 File**].

In Kenya, the overall percentage of antimalarial prescriptions from qualified sources in children under five for recent malarial fever was high in all regions, with the highest percentages observed in Eastern, Western, and Nyanza regions about 100%, and the lowest percentages were observed in Coast (59.4%) region [**Fig 10**]. However, the prevalence of antimalarial drug consumption was highest in Northeast, Central, and Nairobi which were about 100% regions, and lowest in Eastern (15.1%) regions [**S9 Table in S1 File**].

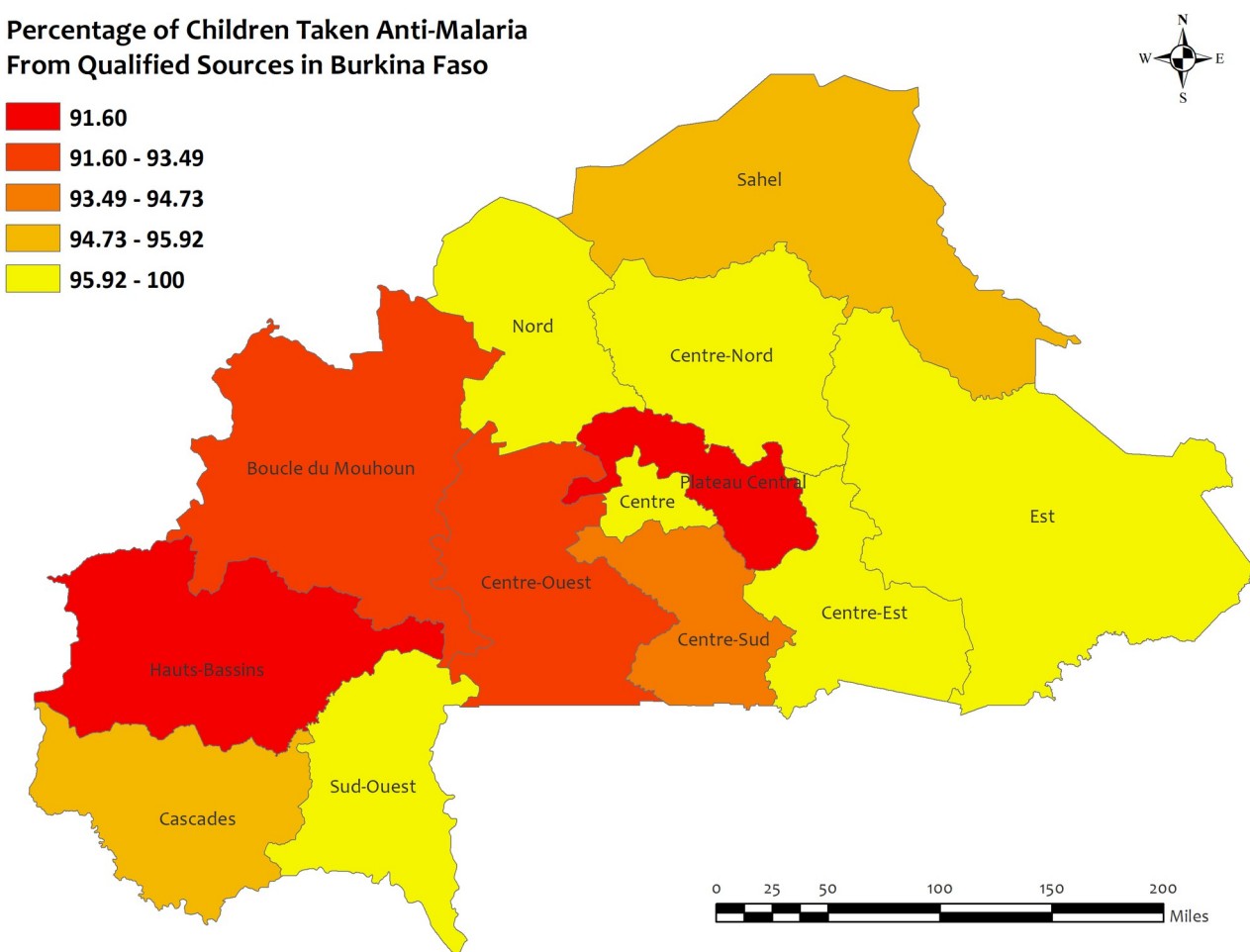

**Fig 5. The prevalence of qualified prescription of antimalarial from qualified sources in Burkina Faso.** Here, the darker shades of red indicate unqualified sources of antimalarial. Basemap data provided by ArcGIS (source:https://hub.arcgis.com/datasets/esri::world-countries-generalized/explore?location=76.272878%2C-132.879863%2C12.88).

In Liberia, the overall percentage of antimalarial prescriptions from qualified sources in children under five for recent malarial fever was high in all regions, with the highest percentages observed in Southeastern B (93.0%) region, while the lowest percentages were observed in Greater Monrovia (61.1%) region [**Fig 11**]. However, the prevalence of antimalarial drug consumption was highest in the South Eastern (100%) region and lowest in the South Central (94.7%) region [**S10 Table in S1 File**].

In Nigeria, the overall percentage of antimalarial prescriptions from qualified sources in children under five for recent malarial fever was high in all regions, with the highest percentages observed in Jigawa, kebbi, Plateau, and Oyo regions at about 100% and the lowest percentages were observed in Rivers (22.7%) regions [**Fig 12**]. However, the prevalence of antimalarial drug consumption was highest in Benue 73.5% region and lowest in Kebbi 0.4% region [**S15 Table in S1 File**].

In Tanzania, the overall percentage of antimalarial prescriptions from qualified sources in children under five for recent malarial fever was high in all regions, with the highest percentages observed in Dodoma, Arusha, Njombe, and Dar Es Salaam regions about 100%, and the lowest percentages observed in Singida (41.6%) regions [**Fig 13**]. However, the prevalence of

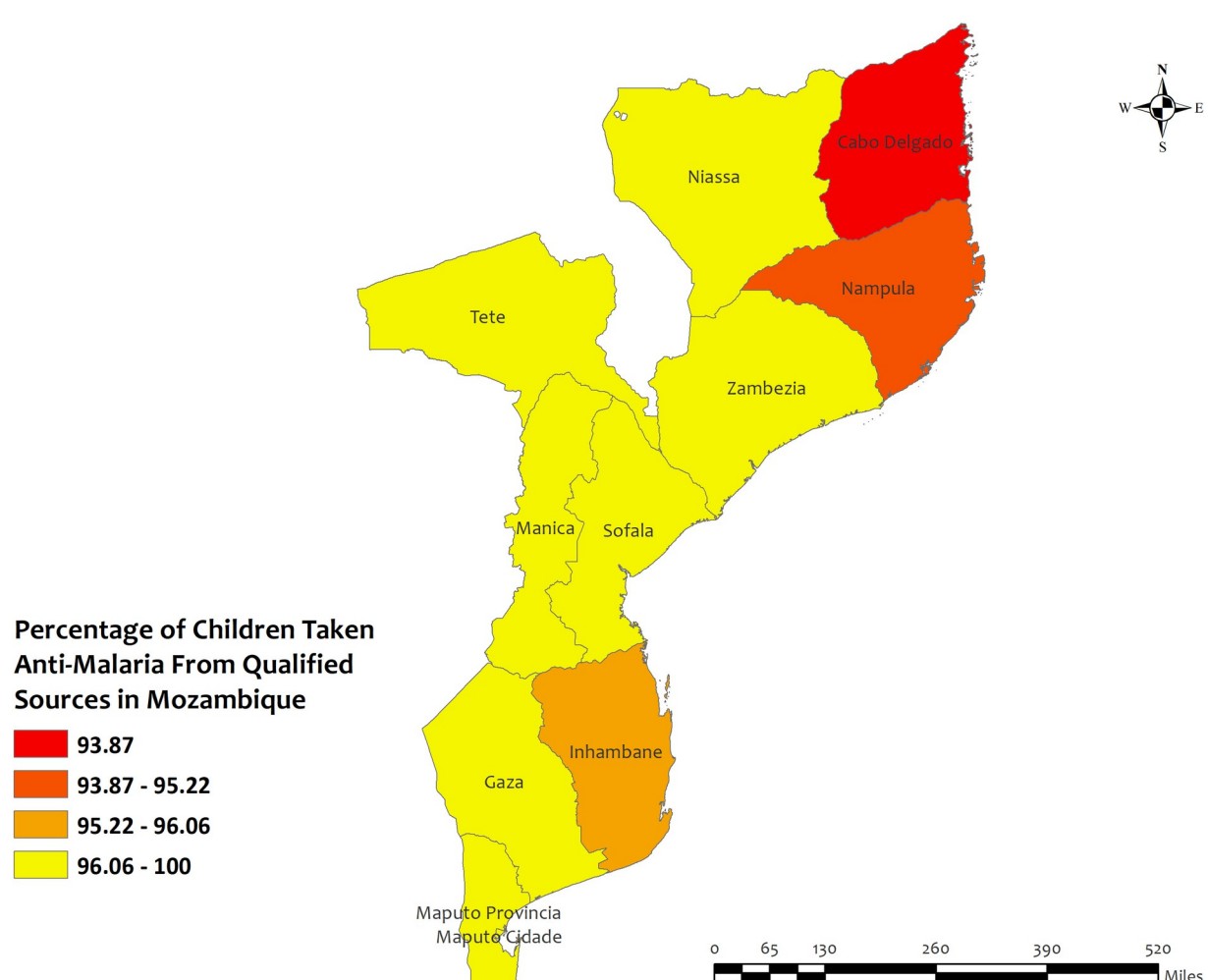

**Fig 6. The prevalence of qualified prescription of antimalarial from qualified sources in Mozambique.** Here, the darker shades of red indicate unqualified sources of antimalarial. Basemap data provided by ArcGIS (source:https://hub.arcgis.com/datasets/esri::world-countries-generalized/explore?location=76.272878%2C-132.879863%2C12.88).

antimalarial drug consumption was 100%, except for the Pwani (96.9%), Tabora (95.6%), and Shinyanga (97.2%) regions [**S19 Table in S1 File**].

## Descriptives statistics for explanatory variables

**Table 2** shows the descriptive statistics for several explanatory factors and antimalarial drugs taken from qualified sources for recent malarial fever in children under five.

About 86.9% of children who are male have taken antimalarial whereas 87.3% of children who are female have taken antimalarial Drugs taken from qualified sources for recent malarial fever in children under five. The percentage of antimalarial Prescription from qualified sources for recent malarial fever seems to decrease as the age of children increases (one year; 89.1%, two years; 87.1%, three years; 87.5%, four years; 84.3%, and five years; 84.2%. Children who live in Rural areas received 88.1% of antimalarial prescriptions from qualified sources, compared to urban areas with 84.3%. 82.3% of children whose parents received higher education took antimalarial prescriptions from qualified sources for recent malarial fever, compared to children whose parents have no education with 90.8%. The percentage of antimalarial

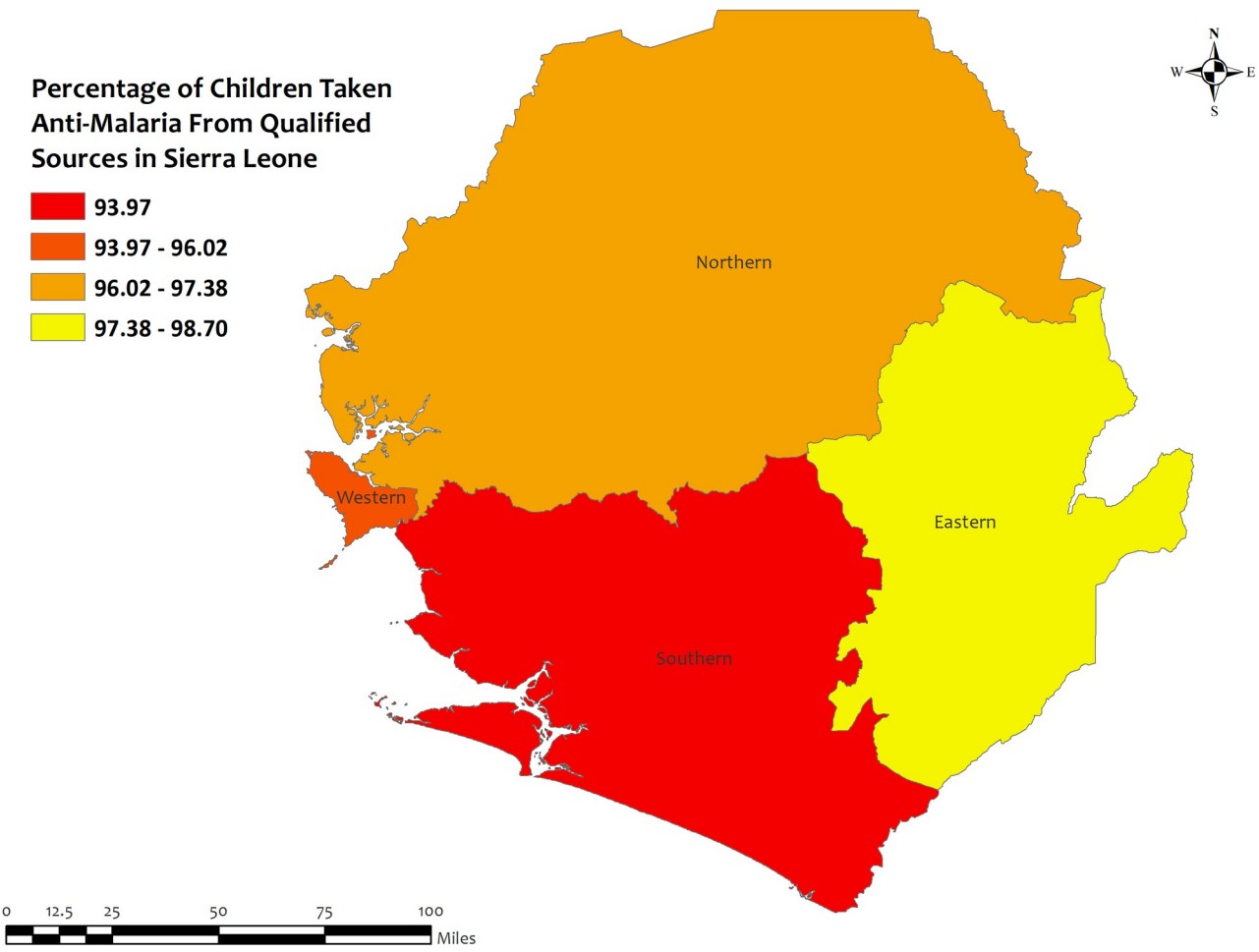

**Fig 7. The prevalence of qualified prescription of antimalarial from qualified sources in Sierra Leone.** Here, the darker shades of red indicate unqualified sources of antimalarial. Basemap data provided by ArcGIS (source:https://hub.arcgis.com/datasets/esri::world-countries-generalized/explore?location=76.272878%2C-132.879863%2C12.88).

prescriptions from qualified sources for the number of children under five in the household are One; 86.0%, two; 89.0%, three; 85.1%, four; 85.0%, and five; 88.4% respectively. According to the wealth index of the child's family, the percentage of taking antimalarial from qualified sources for recent malarial fever seems to decrease from poorest to richest order (90.9% for poorest, 87.2% for poorer, 86.8% for middle, 84.4% for richer and 83.7% for richest [**Table 2**].

## Discussion

Few fevers are treated with effective antimalarials within 24 hours of the beginning of symptoms, even though prompt availability of effective malaria treatment is essential to the success of malaria control efforts globally [28]. The main barriers to reducing malaria mortality may be found in the ineffective testing of malarial fever in children and generally inefficient malaria care, the administration of antimalarial medicines without first performing blood tests for the disease, the administration of more antimalarial medicines than is suggested, or the administration of delayed antimalarial treatment [29]. Our study found that overall, the percentage of antimalarial prescriptions for recent malarial fever from qualified sources in children under

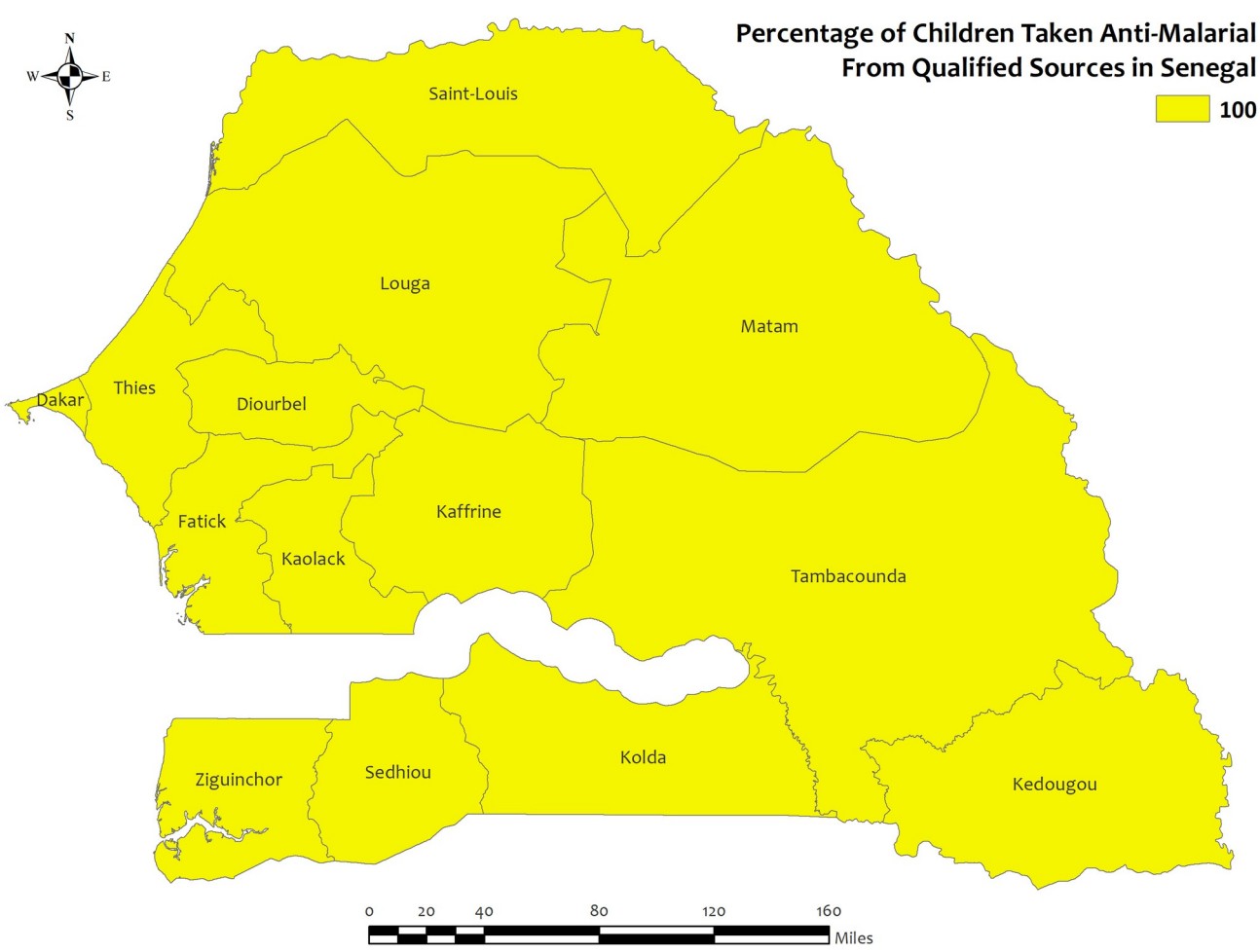

**Fig 8. The prevalence of qualified prescription of antimalarial from qualified sources in Senegal.** Here, the darker shades of red indicate unqualified sources of antimalarial. Basemap data provided by ArcGIS (source:https://hub.arcgis.com/datasets/esri::world-countries-generalized/explore?location= 76.272878%2C-132.879863%2C12.88).

five was very high (above 85 percent) across the LMICs. However, in several LMICs (Guinea, Mali, Nigeria, Kenya, and Senegal), the percentage of antimalarial drug consumption was low.

Moreover, several studies that found various factors that influence high rates of qualified prescriptions are insufficient diagnostic services, a lack of antibiotic guidelines, difficulty in monitoring patient progress, unsatisfactory intensive care facilities in rural areas, patient demand for immediate relief, apparent patient anticipations from prior prescriptions, using up production, and apprehension about losing patients to competition [30–33].

The standard of malaria treatment is still subpar and varies greatly among endemic LMICs. There are several reasons for unqualified prescriptions of antimalarial for recent fever in some of the LMICs (Tanzania, Nigeria, and Ghana). Medications are frequently administered regardless of the findings of a malaria test, indicating that presumptive diagnosis is still frequently used in situations of probable malaria instead of the WHO's advice to "test and treat" [34]. The major causes of inappropriate prescriptions of antimalarial medications, according to a systematic review in SSA regions, include limited diagnostic competence, dependence on clinical symptoms too much, noncompliance with treatment recommendations, and a lack of access to health facilities [35].

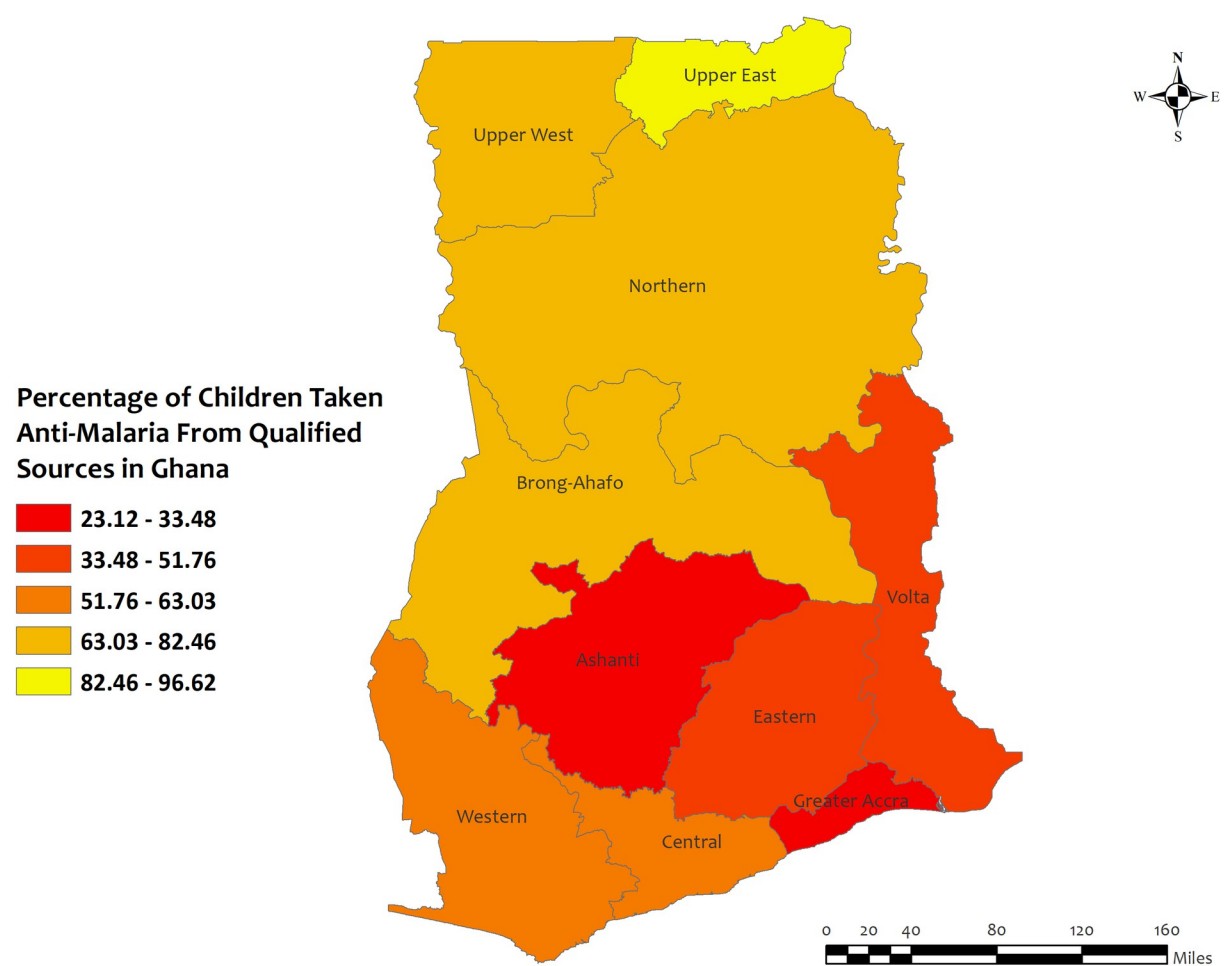

**Fig 9. The prevalence of qualified prescription of antimalarial from qualified sources in Ghana.** Here, the darker shades of red indicate unqualified sources of antimalarial. Basemap data provided by ArcGIS (source:https://hub.arcgis.com/datasets/esri::world-countries-generalized/explore?location=76.272878%2C-132.879863%2C12.88).

Furthermore, traditional, and herbal remedies tend to be the preferred alternative to antimalarial medication in areas where malaria is endemic. According to WHO, 80% of people in third world countries rely on herbal remedies to prevent malaria [36]. Moreover, the high antimalarial prescription rates from unqualified sources are caused by inadequate healthcare infrastructure. The healthcare system has several challenges, including limited healthcare facilities and inadequate funds to address healthcare needs. As a result, it is less likely that those in need of antimalarial medications will be able to obtain them from qualified sources [37–39].

There are several factors that cause some variations in antimalarial prescription for recent malarial fever in children under five.

Children in rural areas typically receive a more qualified prescription of antimalarial for recent malarial fever than those in urban areas. There are several reasons for low qualified prescriptions in urban areas. As urban residents are more informed and have a basic understanding of drugs for common childhood illnesses, they purchase medicines for their children without first seeing a physician [40, 41]. Second, there are more pharmacies in cities than in rural regions, which encourages people to buy antibiotics without prescriptions [42]. Lastly, long wait times in metropolitan government hospitals are another reason why individuals

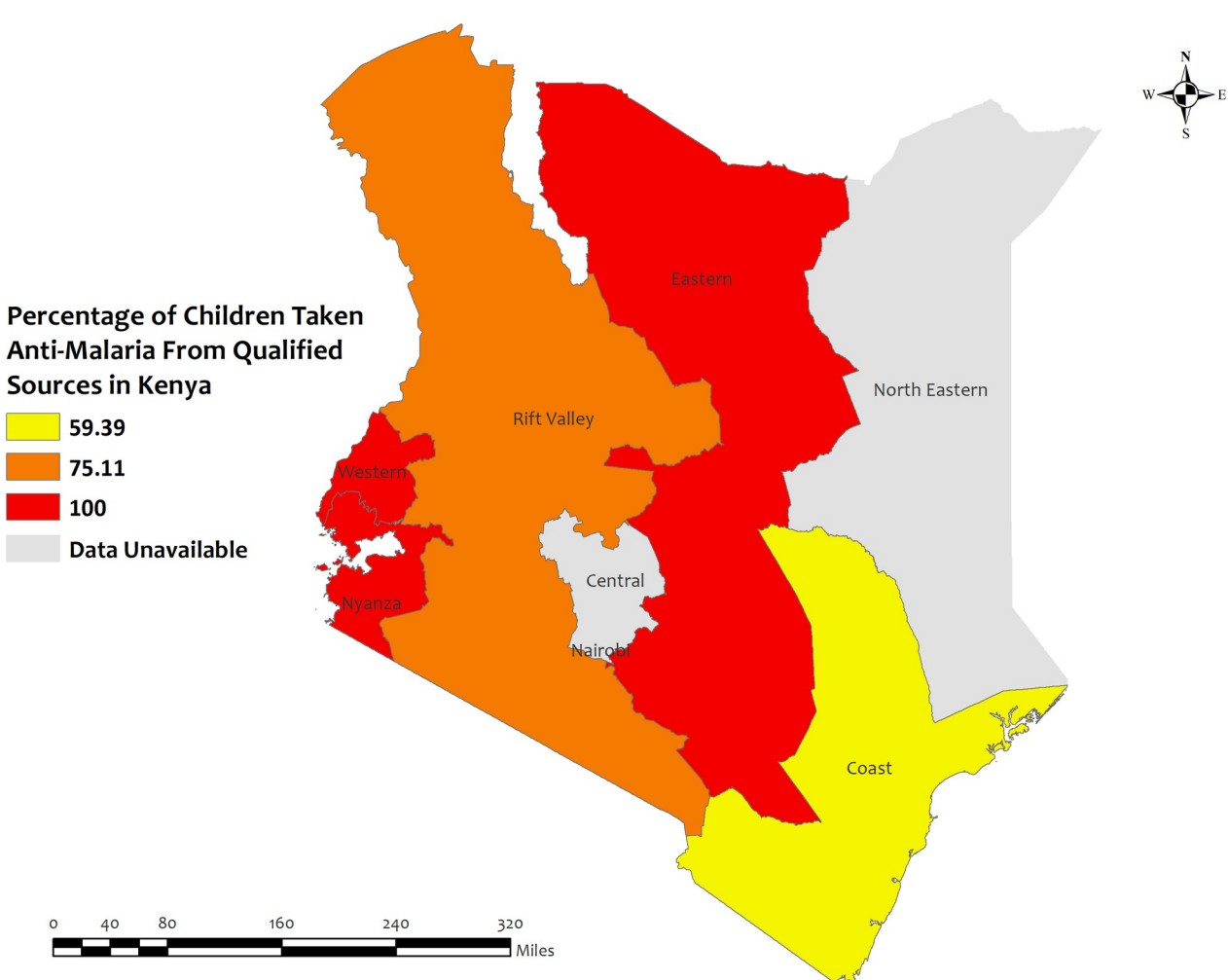

**Fig 10. The prevalence of qualified prescription of antimalarial from qualified sources in Kenya.** Here, the darker shades of red indicate unqualified sources of antimalarial. Basemap data provided by ArcGIS (source:https://hub.arcgis.com/datasets/esri::world-countries-generalized/explore?location=76.272878%2C-132.879863%2C12.88).

avoid seeking treatment from government health providers for frequent and recurrent children's illnesses [43].

Children with mothers who had higher education are likely to get better qualified antimalarial prescriptions for recent malarial fever than children with mothers who have no education. Effective antimalarial usage in young children is linked to caregiver education [44]. In Zambia, children with caregivers who had some educations were less likely to abuse antimalarial drugs [45]. According to Ugandan research, education may have an impact on patients' comprehension of clinic instructions, the quality of their connection with their healthcare practitioner, and their capacity to comprehend visual instructions [46]. Alongside education, the wealth index also affects the rate of antimalarial prescriptions from qualified sources for recent malarial fever in children under five. Children from wealthier families are more likely to get better antimalarial care than those from underprivileged backgrounds. The cause is like the situation in education. Hospital and healthcare facility visits by wealthy people are significantly higher than those by impoverished people.

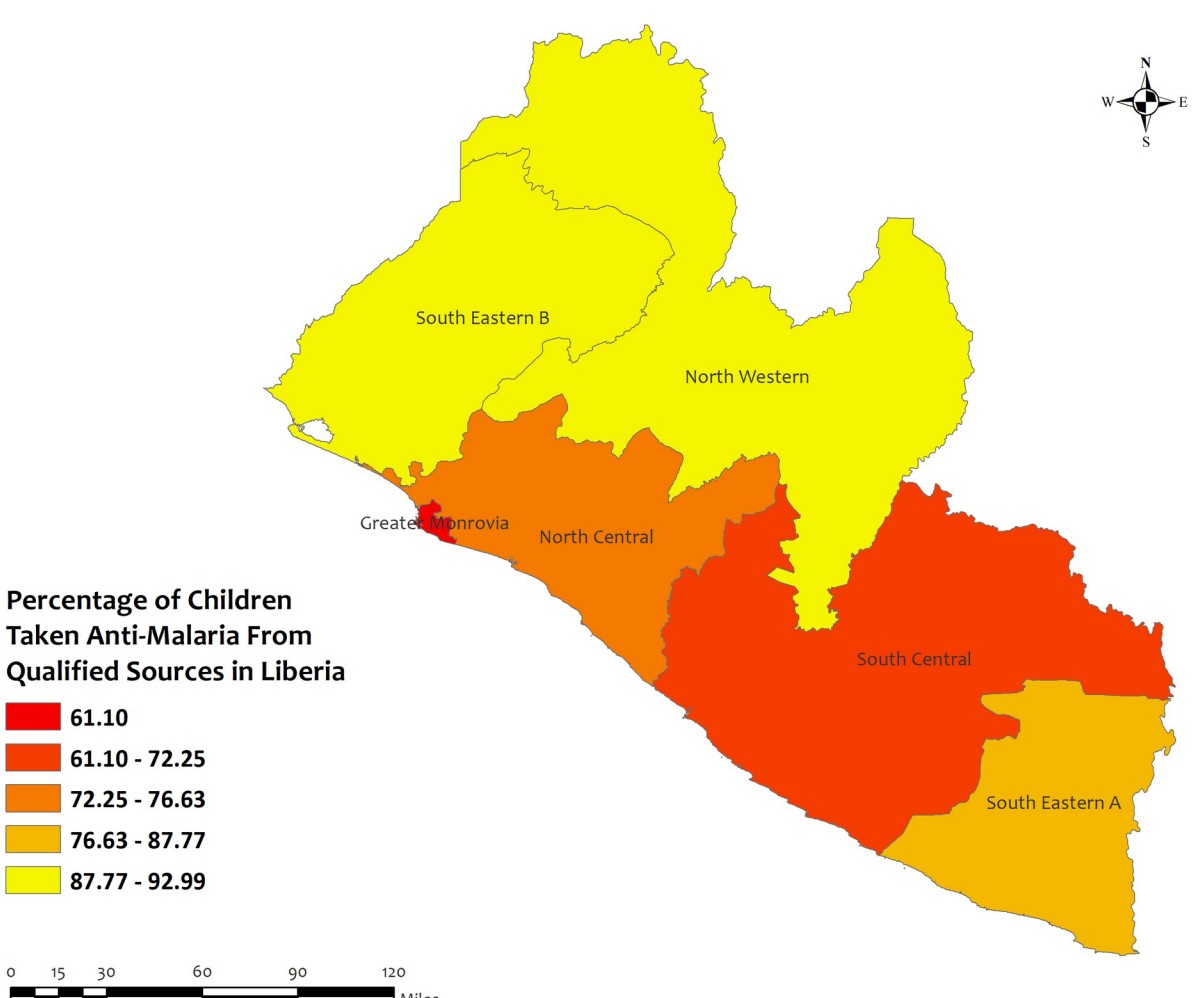

**Fig 11. The prevalence of qualified prescription of antimalarial from qualified sources in Liberia.** Here, the darker shades of red indicate unqualified sources of antimalarial. Basemap data provided by ArcGIS (source:https://hub.arcgis.com/datasets/esri::world-countries-generalized/explore?location=76.272878%2C-132.879863%2C12.88).

Regardless of whether the antimalarials are obtained from qualified sources or not, overprescribing them might lead to the development of antimalarial resistance. Antimalarial resistance can also develop and spread as a result of other reasons, including the abuse of antimalarial medications, poor care, and ineffective preventative strategies [18, 47, 48]. Over-prescription of antimalarial drugs and the emergence of resistance have been linked, according to several studies. Almost 80% of children with fever were given antimalarials, according to a study done in Kenya, and this over-prescription was connected to the emergence of chloroquine resistance [49]. Another study in Tanzania discovered that more than 80% of patients with symptoms similar to malaria were prescribed antimalarials, and this overprescribing was linked to the emergence of resistance to antimalarial prescriptions [50]. Quite apart from the fact that prescribing too many antimalarial drugs might contribute to the establishment and spread of resistance, it is essential to tackle all these causes. To lessen the impact of malaria and stop the emergence of resistant strains, proper antimalarial usage and the execution of preventative measures are crucial [51–53].

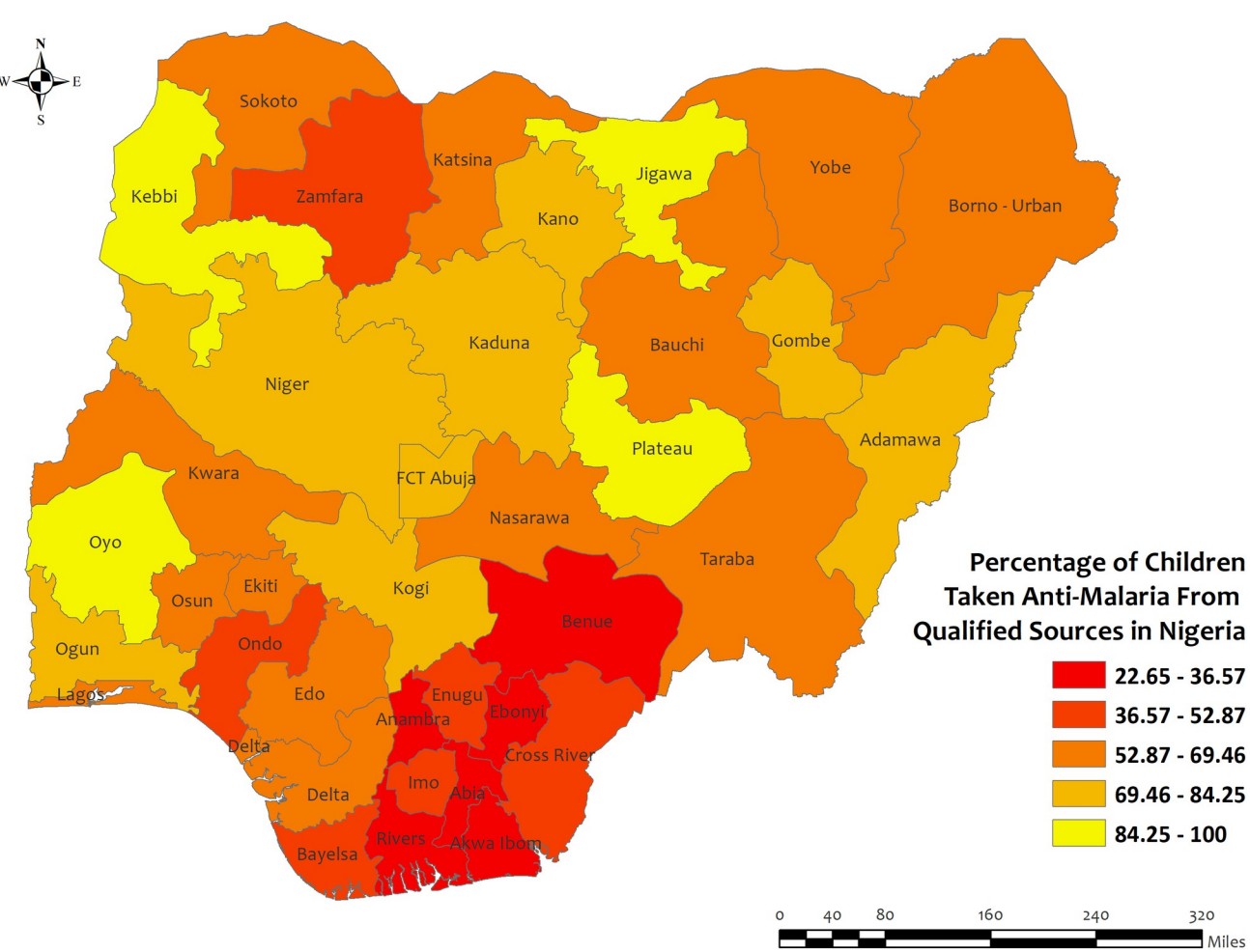

**Fig 12. The prevalence of qualified prescription of antimalarial from qualified sources in Nigeria.** Here, the darker shades of red indicate unqualified sources of antimalarial. Basemap data provided by ArcGIS (source:https://hub.arcgis.com/datasets/esri::world-countries-generalized/explore?location= 76.272878%2C-132.879863%2C12.88).

The high percentages of antimalarial prescriptions for recent malarial fever in children under five are suspected cases. It is essential to explore the existence and adherence to national guidelines for the diagnosis and treatment of malaria. Several studies have highlighted challenges in implementing guidelines, including ineffective testing, inefficient malaria care, and the administration of antimalarials without proper blood tests, potentially contributing to overprescription [28, 29]. A study in Sub-Saharan Africa found that overprescription was linked to limited diagnostic competence, reliance on clinical symptoms, and noncompliance with treatment recommendations [35]. Addressing these issues is crucial to ensuring that antimalarials are prescribed judiciously, avoiding unnecessary use and potential development of resistance [18, 47, 48].

## Recommendations

In Sub-Saharan African countries, it is imperative to implement a comprehensive approach for the effective management of malaria cases. Clear and updated guidelines should be developed and disseminated widely, emphasizing the importance of confirming malaria cases

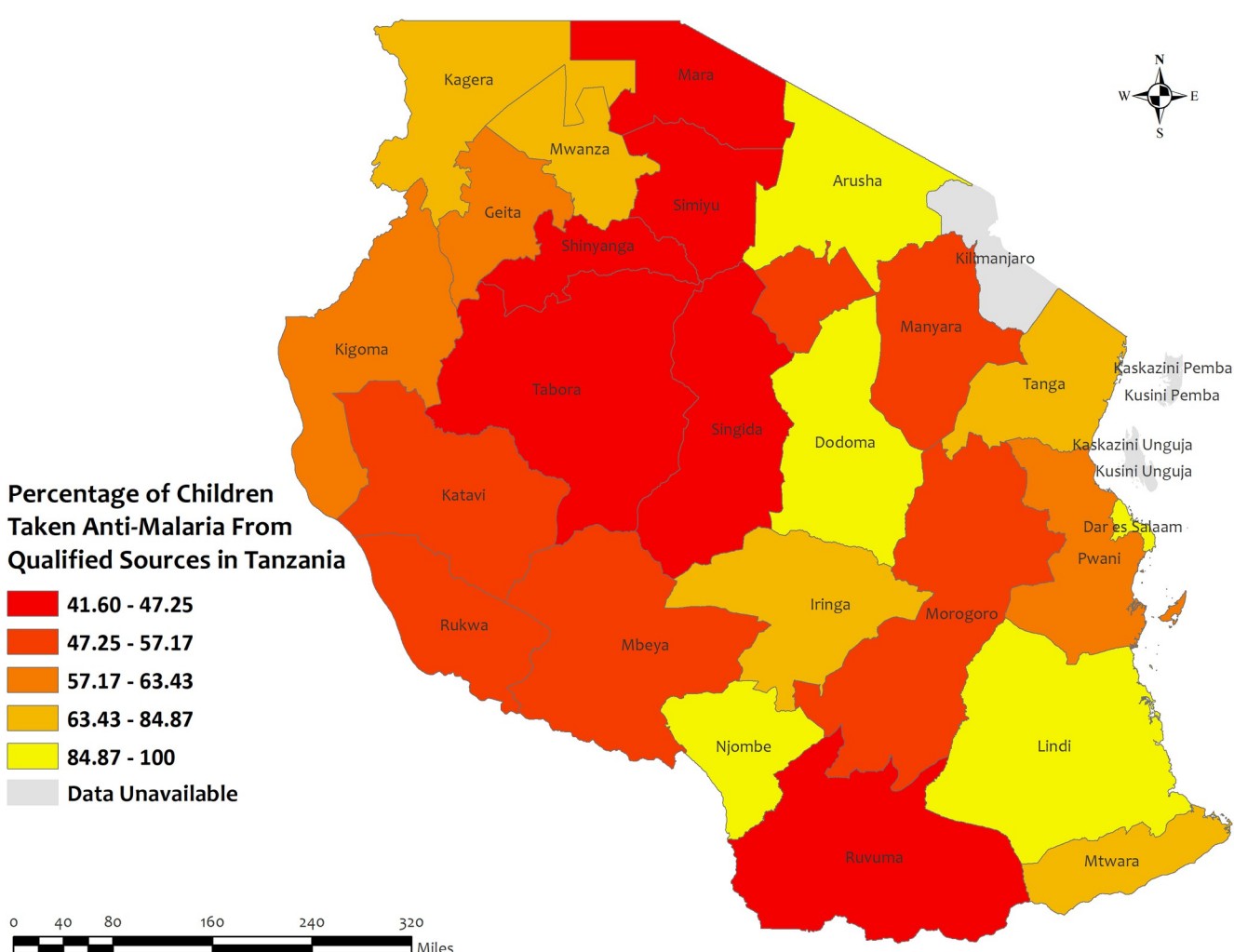

**Fig 13. The prevalence of qualified prescription of antimalarial from qualified sources in Tanzania.** Here, the darker shades of red indicate unqualified sources of antimalarial. Basemap data provided by ArcGIS (source:https://hub.arcgis.com/datasets/esri::world-countries-generalized/explore?location=76. 272878%2C-132.879863%2C12.88).

before prescribing antimalarials, with healthcare professionals receiving adequate training. Rigorous quality assurance measures must be in place to ensure that antimalarial medications are sourced exclusively from qualified and reputable suppliers. Understanding the root causes of malaria prevalence in each country is crucial, requiring comprehensive studies that account for regional variations. Regular assessments should be conducted to gauge the population's knowledge of malaria and the effectiveness of existing control interventions, with tailored health education programs addressing awareness gaps. Communication strategies promoting early utilization of health facilities, the availability of diagnostic tools, and collaboration with traditional practitioners are vital components. Sustainable strategies to counteract non-compliance with treatment regimens and efforts to improve healthcare delivery point accessibility, particularly in rural areas, are essential. Strengthening communication about the nearest health facilities and ensuring a reliable supply chain for diagnostic inputs and medicines are paramount for successful malaria management in the region.

**Table 2. Weighted descriptive statistics of socio-economic variables associated with antimalarial prescriptions from qualified sources for recent malarial fever in the pooled data.**

| Variables | Category | Antimalarial Taken for Recent Fever, Weighted N (%) | Missing percentages (%) |
|---|---|---|---|
| Child's age | One year old | 1198 (89.1) | 7.9 |
| | Two years old | 1807 (87.1) | |
| | Three years old | 1588 (87.5) | |
| | Four years old | 1301 (84.3) | |
| | Five years old | 986.9 (84.2) | |
| Sex of the Child | Male | 3795 (86.9) | 0 |
| | Female | 3716 (87.3) | |
| Type of place of residence | Urban | 1743 (84.3) | 1.4 |
| | Rural | 5674 (88.1) | |
| Highest educational level of Mother's | No education | 2772 (90.8) | 0 |
| | Primary | 3132 (87.5) | |
| | Secondary | 1412 (80.5) | |
| | Higher | 195.4 (82.3) | |
| Number of children under 5 in the household | One | 2483 (86.0) | 0 |
| | Two | 3042 (89.0) | |
| | Three | 1159 (85.1) | |
| | Four | 398.9 (85.0) | |
| | Five or above | 428 (88.4) | |
| Wealth index combined | Poorest | 1911 (90.9) | 0 |
| | Poorer | 1732 (87.2) | |
| | Middle | 1663 (86.8) | |
| | Richer | 1283 (84.4) | |
| | Richest | 922 (83.7) | |

## Strength

To the best of the authors' knowledge, this is the first study on the antimalarial prescription for recent malarial fever in children under five. We were able to explore a comprehensive and detailed picture of antimalarial prescriptions from qualified sources in each of the 19 LMICs with endemic using geospatial analysis. The region-wise GIS map allows local and national officials to easily get a complete view of antimalarial prescriptions.

## Limitations

The results of malaria tests performed on children under the age of five are absent from the MIS databases of most countries. Hence, we considered children under the age of five who have recently had malaria in LMICs. Finally, we focused on children under five. However, there is a need for more studies in areas where malaria is endemic, including studies for all age groups.

## Conclusion

According to our study, even though LMICs were getting high antimalarial prescriptions from qualified sources, certain LMICs were getting high antimalarial prescriptions from unqualified sources for recent malarial fever (Tanzania, Nigeria, and Ghana). Children under the age of five who live in rural areas, have mothers with better education levels, and have more affluent parents, are more frequently to receive antimalarial drugs from qualified sources for malaria treatment. The study focuses on the use of antimalarial prescriptions for recent malarial fever

in children under five in LMICs. We found that greater health infrastructure with checks and balances among antimalarial drugs should be implemented. According to the researchers, the goal of this study is to alert the local authorities to this dreadful situation and recommend need-based support services at the individual level. In the future, surveys will be able to follow kids both throughout the cascade and after their treatment is finished to assess the outcomes of the causality of the child's quality of malaria care.

## Supporting information

**S1 Checklist. STROBE (Strengthening the Reporting of Observational Studies in Epidemiology) checklist used to ensure comprehensive and transparent reporting of observational studies.**
(DOC)

**S1 File. Supplementary Materials containing additional figures, tables, and detailed information to complement and support the findings presented in the manuscript.**
(DOCX)

**S1 Dataset. Raw dataset in CSV format named "Dataset_Antimalarial," containing the data used for analysis in the manuscript.**
(CSV)

## Acknowledgments

We gratefully acknowledge the MIS Program for granting access to the LMICs datasets.

## Author Contributions

**Conceptualization:** Md Sabbir Hossain.

**Data curation:** Md Sabbir Hossain.

**Formal analysis:** Md Sabbir Hossain, Talha Sheikh Ahmed.

**Investigation:** Md Jamal Uddin.

**Methodology:** Md Sabbir Hossain, Md Jamal Uddin.

**Project administration:** Md Jamal Uddin.

**Resources:** Md Sabbir Hossain, Md Jamal Uddin.

**Software:** Md Sabbir Hossain, Muhammad Abdul Baker Chowdhury, Md Jamal Uddin.

**Supervision:** Mohammad Anamul Haque, Muhammad Abdul Baker Chowdhury, Md Jamal Uddin.

**Validation:** Muhammad Abdul Baker Chowdhury, Md Jamal Uddin.

**Visualization:** Talha Sheikh Ahmed.

**Writing – original draft:** Md Sabbir Hossain, Talha Sheikh Ahmed, Mohammad Anamul Haque.

**Writing – review & editing:** Md Sabbir Hossain, Talha Sheikh Ahmed, Mohammad Anamul Haque, Muhammad Abdul Baker Chowdhury, Md Jamal Uddin.

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
