## [Decision Letter · Decision Letter 0]

20 Nov 2023

PONE-D-23-27254Prevalence of unqualified sources of antimalarial drug prescription for children under the age of five: a study in 19 low- and middle-income countriesPLOS ONE

Dear Dr. Uddin,

Thank you for submitting your manuscript to PLOS ONE. After careful consideration, we feel that it has merit but does not fully meet PLOS ONE’s publication criteria as it currently stands. Therefore, we invite you to submit a revised version of the manuscript that addresses the points raised during the review process.

**ACADEMIC EDITOR: **

The paper provides useful informations that will improve MIS data collection and analysis and their use to improve malaria control and elimination strategies. However the paper needs to be reworked by taking into account the following comments in addition to reviewer's comments

We look forward to receiving your revised manuscript.

Kind regards,

Sylla Thiam, M.D, MPH

Academic Editor

PLOS ONE

4. We note that Figures 2, 3, 4, 5, 6, 7, 8, 9, 10, 11, and 12 in your submission contain [map/satellite] images which may be copyrighted. All PLOS content is published under the Creative Commons Attribution License (CC BY 4.0), which means that the manuscript, images, and Supporting Information files will be freely available online, and any third party is permitted to access, download, copy, distribute, and use these materials in any way, even commercially, with proper attribution. For these reasons, we cannot publish previously copyrighted maps or satellite images created using proprietary data, such as Google software (Google Maps, Street View, and Earth). For more information, see our copyright guidelines: http://journals.plos.org/plosone/s/licenses-and-copyright.

1. You may seek permission from the original copyright holder of Figures 2, 3, 4, 5, 6, 7, 8, 9, 10, 11, and 12 to publish the content specifically under the CC BY 4.0 license. 

Methods

- Even though the MIS received ethical clearance in the respective countries. It would have been good for the researchers to seek for an ethical clearance from their own institution

- The reasons given by the authors on the absence of information related to « non response » is too light. Because translation could be used to translate documents in english and access to key information.

Results: Suggest to improve the countries map by comparing the malaria patterns and the antimalaria prescriptions maps by country

Discussion: This sentence is repeated twice in two successive paragraphs

« Regulatory bodies in the country lack the necessary power to effectively control drug pricing or

curb the excessive usage of generic medicines. As a result, the availability of generic drugs is

limited, leading to a significant markup in prices »

Reviewers' comments:

Reviewer's Responses to Questions

**Comments to the Author**

1. Is the manuscript technically sound, and do the data support the conclusions?

Reviewer #1: Yes

2. Has the statistical analysis been performed appropriately and rigorously? 

Reviewer #1: Yes

3. Have the authors made all data underlying the findings in their manuscript fully available?

Reviewer #1: Yes

4. Is the manuscript presented in an intelligible fashion and written in standard English?

Reviewer #1: Yes

5. Review Comments to the Author

Reviewer #1: General comments

The authors should edit number in table 1 and Figure 1 and rework the discussion to focus it according the goal of this study. The authors also should make more effort to summarize the sections to reduce the number of pages.

I appreciated this article because of its originality and recommended its publication provided the authors will do the revision according our comments.

Abstract

Background

Antimalarial resistance poses a severe danger to global health. add drug resistance

Findings

About 60% instead 40%

Introduction

Introduction must be more focus with the tittle:

First describe malaria situation around the world emphasizing the part of children under five on malaria cases and death, the strategies of malaria diagnostic and treatment, the availability of the diagnostic tools and antimalarial drug and the drug resistance management plan.

The second section of the introduction must talk about the justification that unqualified treatments is significantly identified and will need to better documented to understand the causes and the possible effects including the threat in malaria drug resistance.

 

Methods

Methods have been well described according data sources, sampling methods and data analyses.

Some inputs to take into account related to the narrative:

Data Gap Analysis on Antimalarial Prescriptions. In the future, surveys will be able to follow kids both throughout the cascade and after their treatment is finished to assess the outcomes of the causality of the child's quality of malaria care. This sentence must be moved to the Conclusion section

Results

Descriptives Analysis

Decimals must be broken into whole numbers in figure 1 and in the narrative.

Total in table 1 for each column doesn’t match with what we have in the text and in figure 1.

Had Fever Recently Antimalarial taken for Malaria Antimalarial taken from Qualified Sources,

Total 29480 11171 7522

In the narrative 29 478 9477 7511

Males make up 54035.8 of them, or 50.9%, while females make up 5229.2, or 48.1%. Not found nor in table 1 neither in Figure 1

Country Descriptives

Page 17. In this section, we presented the region-wise prevalence of five countries with the highest antimalarial prescriptions from qualified sources for recent malarial fever in children under five. This section must be more summarized as it is expected that there would not be a big difference between zones.

Discussion

P 14. The reasons for the high percentages of qualified prescriptions of antimalarial for recent malarial fever in children under five across the LMICs are manifold…….. P 22. Several studies highlight the shortcomings of ostensibly straightforward approaches (such as fever charts or RDTs) to properly manage fevers and drug usage at the community level45,46,47 . To be deleted and focus must in Africa, about malaria prescription and malaria drug consumption in children under five.

Below this section, from ‘’Moreover, several studies that found various factors that influence high rates of qualified prescriptions are insufficient diagnostic services, a lack of antibiotic guidelines, difficulty in monitoring patient progress, unsatisfactory intensive care facilities in rural areas, patient demand for immediate relief, apparent patient anticipations from prior prescriptions, using up production, and apprehension about losing patients to competition’’, this is appreciated to be in the discussion.

The discussion must be reworked to follow the different points:

• Had Fever Recently,

• Antimalarial taken for Malaria

• Antimalarial taken from Qualified Sources

I putsome aspects that would be taken into account in the discussion et recommendations.

Antimalarial taken for Malaria

Discussion

For the high percentages, it will be necessary to ascertain whether they are suspected or confirmed cases.

If suspected, are there national guidelines for the diagnosis and treatment of malaria?

If so, why are they not respected?

If not, what are the constraints for having the guidelines?

If malaria is confirmed, does the low-test positivity rate justify the low percentages of antimalarial drug use?

Recommendations

Make available guidelines for the management of malaria cases and their application

Require confirmation of malaria cases before prescribing antimalarials

Antimalarial taken from Qualified Sources

Discussion

For the low, it will be necessary to try to find the main causes for each country concerned.

Assessment of the level of knowledge of malaria by the population and the control interventions?

Are there communication strategies for early use of health facilities?

Are inputs for malaria diagnosis and treatment available and accessible?

Level of collaboration between traditional practitioners and the Ministry of Health?

Recommendations

Strengthen communication regarding the use of the nearest health facilities. Implement sustainable strategies capable of counteracting non-compliant treatment causes.

Make available and permanent diagnostic inputs and medicines for the treatment of malaria cases.

Making health care delivery points accessible

6. PLOS authors have the option to publish the peer review history of their article (what does this mean?). If published, this will include your full peer review and any attached files.

Reviewer #1: No

---

## [Author Response · Author response to Decision Letter 0]

11 Feb 2024

Review Responses

Review Comments to the Author

Reviewer #1:

 General comments

The authors should edit number in table 1 and Figure 1 and rework the discussion to focus it according to the goal of this study. The authors also should make more effort to summarize the sections to reduce the number of pages.

I appreciated this article because of its originality and recommended its publication provided the authors will do the revision according to our comments.

Response: We express our gratitude to the reviewers for their invaluable feedback aimed at enhancing the quality of our research. In response to their comments, we have made the necessary updates to Table 1 and Figure 1. Additionally, as per their suggestions, the discussion section has undergone a thorough revision to provide a more concise and summarized overview.

Abstract

Background

Antimalarial resistance poses a severe danger to global health. add drug resistance.

Response: We have revised the abstract accordingly on lines: 27 and page: 2.

“Antimalarial drug resistance poses a severe danger to global health.”

Findings

About 60% instead 40%

Response: We have revised the abstract accordingly on lines: 42 and page: 2.

“In several LMICs (Tanzania, Nigeria, and Ghana), a substantial portion of recent antimalarial prescriptions for malaria was taken from unqualified sources (about 60%).”

Introduction 

Introduction must be more focus with the tittle:

First describe malaria situation around the world emphasizing the part of children under five on malaria cases and death, the strategies of malaria diagnostic and treatment, the availability of the diagnostic tools and antimalarial drug and the drug resistance management plan.

The second section of the introduction must talk about the justification that unqualified treatments is significantly identified and will need to better documented to understand the causes and the possible effects including the threat in malaria drug resistance.

Response: We have revised the introduction following the suggestions provided by the reviewer on lines: 57-141 and pages 3-5.

Methods

Methods have been well described according to data sources, sampling methods and data analyses.

Some inputs to take into account related to the narrative:

Data Gap Analysis on Antimalarial Prescriptions. In the future, surveys will be able to follow kids both throughout the cascade and after their treatment is finished to assess the outcomes of the causality of the child's quality of malaria care. This sentence must be moved to the Conclusion section.

Response: Thank you for your comments. We moved the sentence “In the future, surveys will be able to follow kids both throughout the cascade and after their treatment is finished to assess the outcomes of the causality of the child's quality of malaria care.” to the conclusion section.

Results

Descriptives Analysis

Decimals must be broken into whole numbers in figure 1 and in the narrative.

Total in table 1 for each column doesn’t match with what we have in the text and in figure 1.

Had Fever Recently Antimalarial taken for Malaria Antimalarial taken from Qualified Sources,

Total 29480 11171 7522

In the narrative 29 478 9477 7511

Males make up 54035.8 of them, or 50.9%, while females make up 5229.2, or 48.1%. Not found nor in table 1 neither in Figure 1

Response: We updated the text and Figure 1 accordingly on lines:266-272 and pages:10.

Country Descriptives

Page 17. In this section, we presented the region-wise prevalence of five countries with the highest antimalarial prescriptions from qualified sources for recent malarial fever in children under five. This section must be more summarized as it is expected that there would not be a big difference between zones.

Response: We have summarized this section on lines: 293-233 and pages:10-13.

Discussion

P 14. The reasons for the high percentages of qualified prescriptions of antimalarial for recent malarial fever in children under five across the LMICs are manifold…….. P 22. Several studies highlight the shortcomings of ostensibly straightforward approaches (such as fever charts or RDTs) to properly manage fevers and drug usage at the community level45,46,47 . To be deleted and focus must in Africa, about malaria prescription and malaria drug consumption in children under five.

Below this section, from ‘’Moreover, several studies that found various factors that influence high rates of qualified prescriptions are insufficient diagnostic services, a lack of antibiotic guidelines, difficulty in monitoring patient progress, unsatisfactory intensive care facilities in rural areas, patient demand for immediate relief, apparent patient anticipations from prior prescriptions, using up production, and apprehension about losing patients to competition’’, this is appreciated to be in the discussion.

Response: We removed section “P14. The reasons for the high percentages of qualified prescriptions of antimalarial for recent malarial fever in children under five across the LMICs are manifold…….. P 22.”and keep from ‘’Moreover, several studies that found various factors that influence high rates of qualified prescriptions are insufficient diagnostic services, a lack of antibiotic guidelines, difficulty in monitoring patient progress, unsatisfactory intensive care facilities in rural areas, patient demand for immediate relief, apparent patient anticipations from prior prescriptions, using up production, and apprehension about losing patients to competition’’ on lines 

The discussion must be reworked to follow the different points:

• Had Fever Recently,

• Antimalarial taken for Malaria

• Antimalarial taken from Qualified Sources

I putsome aspects that would be taken into account in the discussion et recommendations.

Antimalarial taken for Malaria

Discussion

For the high percentages, it will be necessary to ascertain whether they are suspected or confirmed cases.

If suspected, are there national guidelines for the diagnosis and treatment of malaria?

If so, why are they not respected?

If not, what are the constraints for having the guidelines?

If malaria is confirmed, does the low-test positivity rate justify the low percentages of antimalarial drug use?

Response: We have added a section in the discussion as suggested by the reviewers on lines:515-524 and pages 19-20.

“The high percentages of antimalarial prescriptions for recent malarial fever in children under five are suspected cases. It is essential to explore the existence and adherence to national guidelines for the diagnosis and treatment of malaria. Several studies have highlighted challenges in implementing guidelines, including ineffective testing, inefficient malaria care, and the administration of antimalarials without proper blood tests, potentially contributing to overprescription28-29. A study in Sub-Saharan Africa found that overprescription was linked to limited diagnostic competence, reliance on clinical symptoms, and noncompliance with treatment recommendations34. Addressing these issues is crucial to ensuring that antimalarials are prescribed judiciously, avoiding unnecessary use and potential development of resistance 18,46-47.” 

Recommendations

Make available guidelines for the management of malaria cases and their application

Require confirmation of malaria cases before prescribing antimalarials

Antimalarial taken from Qualified Sources

Discussion

For the low, it will be necessary to try to find the main causes for each country concerned.

Assessment of the level of knowledge of malaria by the population and the control interventions?

Are there communication strategies for early use of health facilities?

Are inputs for malaria diagnosis and treatment available and accessible?

Level of collaboration between traditional practitioners and the Ministry of Health?

Recommendations

Strengthen communication regarding the use of the nearest health facilities. Implement sustainable strategies capable of counteracting non-compliant treatment causes.

Make available and permanent diagnostic inputs and medicines for the treatment of malaria cases.

Making health care delivery points accessible.

Response: We adjusted the recommendation section based on the reviewer's feedback on lines:526-554 and pages: 20-21 

“In Sub-Saharan African countries, it is imperative to implement a comprehensive approach for the effective management of malaria cases. Clear and updated guidelines should be developed and disseminated widely, emphasizing the importance of confirming malaria cases before prescribing antimalarials, with healthcare professionals receiving adequate training. Rigorous quality assurance measures must be in place to ensure that antimalarial medications are sourced exclusively from qualified and reputable suppliers. Understanding the root causes of malaria prevalence in each country is crucial, requiring comprehensive studies that account for regional variations. Regular assessments should be conducted to gauge the population's knowledge of malaria and the effectiveness of existing control interventions, with tailored health education programs addressing awareness gaps. Communication strategies promoting early utilization of health facilities, the availability of diagnostic tools, and collaboration with traditional practitioners are vital components. Sustainable strategies to counteract non-compliance with treatment regimens and efforts to improve healthcare delivery point accessibility, particularly in rural areas, are essential. Strengthening communication about the nearest health facilities and ensuring a reliable supply chain for diagnostic inputs and medicines are paramount for successful malaria management in the region.”

---

## [Decision Letter · Decision Letter 1]

27 Feb 2024

Prevalence of unqualified sources of antimalarial drug prescription for children under the age of five: a study in 19 low- and middle-income countries

PONE-D-23-27254R1

Dear Dr. Uddin

We’re pleased to inform you that your manuscript has been judged scientifically suitable for publication and will be formally accepted for publication once it meets all outstanding technical requirements.

Kind regards,

Sylla Thiam, M.D, MPH

Academic Editor

PLOS ONE

Additional Editor Comments (optional):

Reviewers' comments:

Reviewer's Responses to Questions

**Comments to the Author**

1. If the authors have adequately addressed your comments raised in a previous round of review and you feel that this manuscript is now acceptable for publication, you may indicate that here to bypass the “Comments to the Author” section, enter your conflict of interest statement in the “Confidential to Editor” section, and submit your "Accept" recommendation.

Reviewer #1: All comments have been addressed

2. Is the manuscript technically sound, and do the data support the conclusions?

Reviewer #1: Yes

3. Has the statistical analysis been performed appropriately and rigorously? 

Reviewer #1: I Don't Know

4. Have the authors made all data underlying the findings in their manuscript fully available?

Reviewer #1: Yes

5. Is the manuscript presented in an intelligible fashion and written in standard English?

Reviewer #1: Yes

6. Review Comments to the Author

Reviewer #1: I confirm that all comments were taken account by the authors . My only additional comment is related to the fig 13 which must be moved as fig 2. Also in this figure the authors have to add the number of male and female.

7. PLOS authors have the option to publish the peer review history of their article (what does this mean?). If published, this will include your full peer review and any attached files.

Reviewer #1: **Yes: **Abdoulaye Diop

---

## [Editor Report · Acceptance letter]

8 Mar 2024

PONE-D-23-27254R1 

PLOS ONE

Dear Dr. Uddin, 

I'm pleased to inform you that your manuscript has been deemed suitable for publication in PLOS ONE. Congratulations! Your manuscript is now being handed over to our production team.

Kind regards, 

on behalf of

Dr. Sylla Thiam 

Academic Editor

PLOS ONE